# Structure-specific rigid dose accumulation dosimetric analysis of ablative stereotactic MRI-guided adaptive radiation therapy in ultracentral lung lesions

## Abstract

**Background** Definitive local therapy with stereotactic ablative radiation therapy (SABR) for ultracentral lung lesions is associated with a high risk of toxicity, including treatment related death. Stereotactic MR-guided adaptive radiation therapy (SMART) can overcome many of the challenges associated with SABR treatment of ultracentral lesions.

**Methods** We retrospectively identified 14 consecutive patients who received SMART to ultracentral lung lesions from 10/2019 to 01/2021. Patients had a median distance from the proximal bronchial tree (PBT) of 0.38 cm. Tumors were most often lung primary (64.3%) and HILUS group A (85.7%). A structure-specific rigid registration approach was used for cumulative dose analysis. Kaplan-Meier log-rank analysis was used for clinical outcome data and the Wilcoxon Signed Rank test was used for dosimetric data.

**Results** Here we show that SMART dosimetric improvements in favor of delivered plans over predicted non-adapted plans for PBT, with improvements in proximal bronchial tree DMax of 5.7 Gy ($p = 0.002$) and gross tumor 100% prescription coverage of 7.3% ($p = 0.002$). The mean estimated follow-up is 17.2 months and 2-year local control and local failure free survival rates are 92.9% and 85.7%, respectively. There are no grade $\geq 3$ toxicities.

**Conclusions** SMART has dosimetric advantages and excellent clinical outcomes for ultracentral lung tumors. Daily plan adaptation reliably improves target coverage while simultaneously reducing doses to the proximal airways. These results further characterize the therapeutic window improvements for SMART. Structure-specific rigid dose accumulation dosimetric analysis provides insights that elucidate the dosimetric advantages of SMART more so than per fractional analysis alone.

## Plain language summary

Stereotactic MR-guided Adaptive Radiation Therapy (SMART) is a type of radiation therapy for cancer. With SMART, treatment can be adapted based on daily changes in the body seen via imaging. SMART can safely deliver radiation to lung tumors near the center of the body which are risky to treat, due to potential damage to nearby organs. We looked at 14 patients who received SMART to determine how much changing the radiation plan each day improved our ability to safely deliver high doses. We found that SMART not only improved our ability to cover the entirety of the tumor with the dose originally intended, but also reduced dose to nearby organs. Treatment resulted in excellent control of the tumor with few side effects. SMART shows promise for safer and more effective treatment for lung tumors in this part of the body.

Stereotactic ablative body radiation therapy (SABR) is a common modality applied in the treatment of both malignant primary and metastatic thoracic tumors. Survival and local disease control outcomes in early-stage non-small cell lung cancer (NSCLC) have been shown to improve when the delivered biologically effective dose of base 10 ($BED_{10}$) is $\geq 100$ Gy[1–3]. However, excess toxicity associated with SABR is often seen in central tumors[4], which are tumors located close to the radiosensitive structures of the mediastinum[5,6]. This excess toxicity risk appears to be greatest in ultracentral lesions[7,8]. The definition of what constitutes an ultracentral thoracic lesion varies[7,9–11]. Numerous strategies have been employed to minimize these increased toxicity risks, including more moderately hypofractionated radiation therapy (RT)[12,13]. Despite lengthening the course of

✉e-mail: john.bryant@moffitt.org; Stephen.Rosenberg@moffitt.org

RT, concerns remain regarding treatment-related toxicity and balancing local control[9].

Both SABR and moderately hypofractionated RT suffer from the same pitfalls related to uncertainties in treatment setup, inter- and intra-fractional organs at risk (OAR) target movement, and gating uncertainties. A common strategy employed to improve target coverage is to use a 4-dimensional computed tomography (4DCT) scan. This strategy can account for intra-fractional movement of the target by including the entire extent of the target's motion during the free breathing cycle into an internal gross tumor volume (IGTV)[14]. However, it also increases the total irradiated volume and dose to nearby OAR. The closer these lesions are to central mediastinal structures, the more challenging it becomes to protect the critical central OAR from elevated radiation doses due to the increased volumes of the IGTVs. SABR is associated with a high rate of treatment-related death in ultracentral lesions[7]. However, the magnetic resonance image (MRI)-guided linear accelerator (MRL) is particularly well suited to addressing the challenges of ultracentral RT.

The MRL has emerged as a state-of-the-art linear accelerator capable of delivering accurate radiation therapy[15–17] for various types of solid cancers[18–25]. MRLs enable safe dose escalation to targets that are nearby critical radiosensitive OAR by utilizing stereotactic MR-guided adaptive radiation therapy (SMART). SMART is an advanced RT modality that combines SABR with the daily plan adaptation and real-time internal anatomy-based beam gating potential of the MRL[11,18,19,21,22,26–28]. SMART increases the therapeutic ratio of SABR for central and ultracentral tumors by eliminating the increased volumes associated with IGTVs[29] by ensuring accurate delivery to the target via real-time direct tumor localization via cine MRI with gating[29–31] and allowing for daily treatment plan adaptation to account for interfractional anatomic shifts and daily setup uncertainties in order to decrease OAR dose, increase target coverage, or both[29,32]. These abilities allow for precise dose delivery to the tumor while simultaneously minimizing radiation exposure to surrounding OAR, thereby potentially increasing disease control and reducing treatment-related toxicities[11,19,31,33].

Although SMART has been associated with a favorable toxicity profile and improvements in target coverage for ultracentral tumors[11,29–34], accurate measurement of the cumulative doses delivered is needed to better understand the true potential that SMART offers these patients. While deformable image registration (DIR) has been used in some prior MR-guided adaptive RT (MRgART) analyses to estimate dose accumulation, it has several limitations that can lead to inaccuracies in dose calculation[35–37]. The local structure-specific rigid image registration (RIR) proposed within this work attempts to address these challenges by avoiding deformation and isolating cumulative dose analysis to a single structure at a time[38].

To the best of our knowledge, we present one of the largest cohorts of ultracentral lung tumor patients to date treated with SMART and the first to demonstrate the dosimetric differences in cumulative delivered versus predicted plans generated with a local structure-specific RIR technique. We hypothesized that plan adaptation to the daily anatomy with SMART would result in improvements of cumulative doses delivered to the tumor while simultaneously reducing doses to the proximal bronchial tree (PBT). Here, we show that SMART dosimetric improvements in favor of delivered plans over predicted non-adapted plans for PBT, with improvements in proximal bronchial tree DMax of 5.7 Gy ($p = 0.002$) and gross tumor 100% pre-scription coverage of 7.3% ($p = 0.002$).

## Methods
### Patient eligibility
After obtaining the H. Lee Moffitt Cancer Center and Research Institute institutional review board approval (#20383), 14 consecutive patients who received ablative SMART (i.e., $BED_{10} > 100$ Gy) for ultracentral lesions between October 2019 and January 2021 were identified. Informed consent was not required as this is a retrospective cohort study. Ultracentral lesions were defined as per the Nordic-HILUS[7] group A and B criteria: ultracentral group A lesions (i.e., gross tumor volume [GTV]) were defined as being ≤1 cm from the main bronchus and trachea and group B lesions were

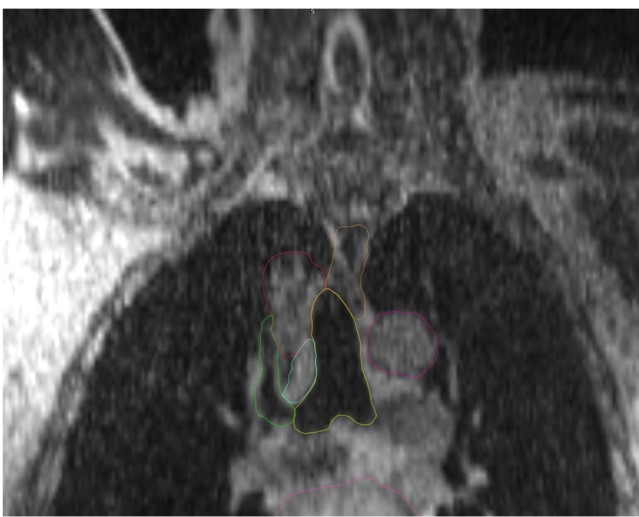

**Fig. 1 | Coronal view of a TRUFI sequence used to contour targets and OAR.** The gross target volume (GTV), trachea, esophagus, right upper lobe bronchi, azygous vein, aorta, and heart are contoured in red, yellow, orange, green, blue, magenta, and pink, respectively.

defined as being ≤1 cm from intermediate and lobar bronchi. Patients with primary lung cancers and metastatic lesions to the lung were included.

### SMART simulation and planning
Patients underwent MRI simulation on the 0.35 T MRIdian System (ViewRay Inc., Mountain View, CA) along with CT simulation for tissue density information. MRI and CT scans were each done in the same supine position using a deep inspiratory breath hold (DIBH) technique. During MRI simulation, a true fast imaging with a balanced steady-state free precession (TRUFI) sequence was used to define the targets and OAR during treatment planning (Fig. 1). An isotropic 3 mm expansion from the GTV was utilized to create the planning target volume (PTV). The structure used for real-time gating was a region with appreciable contrast from surrounding regions and completely encompassed by the PTV to ensure appropriate dosimetric coverage. A gating boundary structure was generated from this with a 3 mm isotropic expansion to match GTV to PTV margin. A percentage excursion threshold of the tracking structure outside of the boundary structure was typically set at <5% to trigger beam-on. Due to the MRL's ability to directly gate the beam during treatment, IGTVs were not necessary, allowing for smaller treated volumes. All treatment planning was performed on the ViewRay Treatment Planning System (TPS). Target coverage goals and OAR constraints are summarized in Supplementary Table 1. All targets were prescribed to 6000 cGy in eight fractions (i.e., $BED_{10} = 105$)[39], and GTV hotspots of up to 120% of the prescription dose were allowed. Full details regarding SMART simulation and planning have been previously reported[29].

### SMART daily adaptive workflow
Greater detail of the SMART daily adaptive workflow has been previously reported[29]. In brief, daily setup MRIs included the anatomy of the day (i.e., target and the entirety of corresponding surrounding patient anatomy). The daily target volumes were rigidly registered to the reference (simulation) MRI. The OAR was then segmented on the daily MRI. The GTV was also edited at this time if needed, but this is less common during an eight-fraction regimen due to a relative lack of changes in tumor geometry over this period. Either the original plan or an adapted plan from one of the previous fractions was recalculated on the anatomy of the day and evaluated for meeting the critical constraints (Supplementary Table 1). If no violations were present, and both the dose-volume histograms and isodose distributions were satisfactory to the treating physician, the plan was then used for treatment. In theory, those plans could be further improved through adaptation. In practice, the time-compressed nature of adaptive treatments, vendor-

specific workflow, and the lack of readily available composite plan evaluation tools (e.g., like the evaluation tools described by Gintz et al.[40]) pointed us to adopting an institutional protocol whereby no changes were made to a plan if it met all the target coverage and OAR sparing constraints. If any violations occurred, the plan was reoptimized (i.e., adapted). The options included either weight optimization or full optimization of the fixed-gantry segmented intensity-modulated plans. Weight optimization is a simpler form of adaptation that alters the weight of the base plan multileaf collimator (MLC) segments without changing their shape. Full optimization includes changes in segment weights, numbers, and shapes based on either original or modified optimization objectives. However, the gantry angles do not change with either option. Plan adaptation based on target coverage was made at the treating physician's discretion. The selected plan was delivered with real-time tumor tracking and gating.

## Per fraction and cumulative plans

Critical structures and associated dosimetric metrics were selected prior to data collection based on estimated clinical significance. Those regions of interest (ROI) were the GTV, PTV, and PBT (i.e., the closest bronchial airway structure to the tumor). These metrics were collected for the base plan (i.e., an initial plan developed from simulation), each delivered fraction plan (i.e., the adapted and non-adapted plans associated with each fraction of therapy), and predicted fraction plans (i.e., the base plan recomputed on the daily dataset at each fraction of therapy). Thus, there was only one base plan per patient; however, each patient had eight delivered and predicted plans. The delivered and predicted plans were compared against each other on both a per fraction analysis and cumulative dosimetric analysis. Per fraction analysis consisted of a direct comparison of every delivered plan to its corresponding predicted plan (e.g., the delivered plan for fraction #1 was only compared to the predicted plan for fraction #1). Cumulative plans were generated using structure-specific RIR. RIR was performed twice: first by aligning to the target (i.e., the GTV) and then again by aligning to the PBT. Thus, each patient had two delivered cumulative plans and two predicted cumulative plans: one for the target and one for the PBT. The cumulative delivered plan for each patient was only compared to the corresponding cumulative predicted plan that focused on the same ROI (i.e., the GTV or PBT). In summary, the RIR-based process resulted in four cumulative treatment plans per patient used: (1) delivered cumulative plan focused on the target, (2) delivered cumulative plan focused on the PBT, (3) predicted cumulative pan for the target, and (4) predicted cumulative plans for the PBT.

## Base plans

Many of the base plans were never actually delivered to the patient because of the adaptive workflow being triggered during every fraction of therapy. However, these plans served as a baseline for comparison with delivered and predicted plans as they were developed without time constraints of online adaptation and should provide a close approximation of what dosimetric quality is achievable given each patient's unique anatomy.

## Delivered plans

The delivered plans were calculated on the anatomy of the day (i.e., the daily MRI setup scan with the original CT deformably registered to it for electron density information) and represented the actual treatment course of radiation therapy. Every patient had eight unique delivered plans. This is caused by slight deviations due to interfractional anatomic shifts and setup differences, even if the same nominal plan was delivered for multiple fractions. The delivered plans consisted of both adapted and non-adapted plans. Non-adapted plans were fractions when the base plan was suitable to be delivered, and the adaptive workflow was not triggered. For per fraction analyses, the dosimetric parameters were collected directly from the MRIdian treatment planning system (TPS).

## Predicted plans

The predicted plan represented the base plan fluence projected and calculated on the anatomy of the day for every fraction. These predicted plans

were used at the beginning of each session to determine if the adaptive workflow should be triggered. Each patient had eight unique predicted plans. If the base plan was acceptable to use on the anatomy of the day then both the delivered and predicted plans were the same for that fraction, although this was a rare occurrence. The dosimetric parameters for the predicted plans were collected directly from the TPS.

Predicted plans were not explicitly stored within the TPS like the delivered plans and needed to be re-generated by using the base plan on the anatomy of the day. The system has a feature that automatically regenerates these plans. Occasionally, this reconstruction failed due to code imperfections. This was rare and never occurred more than once per patient. When that happened, a comparison between the delivered and predicted plans for that fraction could not be performed.

## Cumulative delivered plans

To generate cumulative delivered plans, the base plan and all delivered plans were exported from the TPS to the Mirada image management system (Mirada Medical Ltd., Oxford, UK). Two cumulative delivered plans were generated for every patient using two versions of structure-specific RIR, one focusing on the GTV and another on the nearest (to the GTV) part of the PBT. The base plan MRI was used as the locus for these RIRs, and the base plan structure set was then used to collect the dosimetric parameters from the generated predicted cumulative plan. It is important to note that the base plan dose was not used to generate any of these delivered plans except for the fractions when the base plan was delivered. RIRs were performed by a single physician using an estimated and manual iterative process. GTVs were aligned to the tumor centroid, and PBTs were aligned to the surface closest to the GTV.

To generate the cumulative plans, every fraction plan was given a weight of 1/8th. Only the dosimetric parameters related to the target (i.e., the GTV and PTV) were collected from the cumulative predicted plan where RIR was performed on the GTV, and only those dosimetric parameters related to the PBT were collected from the cumulative predicted plan where RIR was performed on the PBT.

## Cumulative predicted plans

The cumulative predicted plans were generated in the same manner as the cumulative delivered plans and each patient had a cumulative plan generated for the target and PBT. The only important difference is that there were rare occurrences where a predicted fraction plan was unable to be generated (as stated above); thus, the cumulative plans generated for that patient were generated by assigned a weight of 1/7th per plan to generate the best approximation given incomplete treatment data.

## Follow-up

Patients were followed up regularly after treatment to collect clinical data. Surveillance imaging with CT or positron emission tomography (PET) was performed every three to 6 months. Toxicities were assessed using Common Terminology Criteria for Adverse Events (CTCAE) v.5.0. Acute toxicities were defined from the start of treatment to less than 90 days post-treatment. Late toxicities were defined as greater than or equal to 90 days from the end of treatment. Data were collected retrospectively for possible treatment-related toxicities and treatment-related outcomes.

## Statistical analysis

All statistical tests were performed with SPSS version 29 software (IBM, Armonk, New York, United States). Follow-up was defined from the end of SMART to the last contact or death. The reverse Kaplan-Meier method was utilized to estimate median follow-up. Clinical outcomes were estimated from the end of SMART to the last follow-up, progression, or death. Local control (LC) was defined as the absence of local failure (LF) as per Response Evaluation Criteria in Solid Tumors (RECIST) confirmed by CT, PET, or biopsy, marginal failure, or involved lobe failure, as per RTOG 0813[4]. Overall survival (OS) was defined as freedom from death of any cause. Progression-free survival (PFS) was defined as the shortest interval of time

## Table 1 | Baseline patient, tumor, and treatment characteristics

| Characteristic | N/median (%/IQR) |
|---|---|
| Age at diagnosis (year) | 65.5 (58.5–73.5) |
| *Gender* | |
| Female | 9 (64.3%) |
| Male | 5 (35.7%) |
| *ECOG performance status* | |
| 0 | 3 (21.4%) |
| 1 | 8 (57.1%) |
| 2 | 3 (21.4%) |
| *Smoking status* | |
| Never | 3 (21.4%) |
| Former | 10 (71.4%) |
| Current | 1 (7.1%) |
| *Non-metastatic primary histology* | |
| Adenocarcinoma | 4 (28.6%) |
| Carcinoid (DIPNECH) | 1 (7.1%) |
| Squamous cell carcinoma | 4 (28.6%) |
| *Metastatic primary histology* | |
| Salivary gland adenoid cystic carcinoma | 1 (7.1%) |
| Cholangiocarcinoma | 1 (7.1%) |
| Colorectal adenocarcinoma | 2 (14.3%) |
| Pulmonary neuroendocrine (mixed large and small cell) | 1 (7.1%) |
| *Non-metastatic lung primary T stage* | |
| T1a | 0 |
| T1b | 1 (7.1%) |
| T1c | 4 (28.6%) |
| T2a | 2 (14.3%) |
| T2b | 2 (14.3%) |
| Maximum pre-treatment tumor diameter (cm) | 2.40 (2.23–3.88) |
| *Lesion lobar location* | |
| Upper lobe, left | 5 (35.7%) |
| Upper lobe, right | 8 (57.1%) |
| Lower lobe, left | 1 (7.1%) |
| *HILUS ultracentral group* | |
| A[a] | 12 (85.7%) |
| B[b] | 2 (14.3%) |
| Distance from trachea or main bronchus (cm) | 0.38 (0.00–0.79) |
| Tumor size (cc) | 17.8 (9.4–38.6) |
| Planning target volume (cc) | 35.0 (18.2–61.4) |
| *Concurrent systemic therapy* | |
| None | 13 (92.9%) |
| Nivolumab | 1 (7.1%) |
| *Prior thorax radiation therapy* | |
| None | 10 (71.4%) |
| Yes, non-overlapping target | 2 (14.3%) |
| Yes, overlapping target | 2 (14.3%) |

*DIPNECH diffuse idiopathic pulmonary neuroendocrine cell hyperplasia, ECOG Eastern Cooperative Oncology Group, IQR interquartile range.*
[a] ≤1 cm from the main bronchus and trachea.
[b] ≤1 cm from intermediate and lobar bronchi.

to either local failure, distant failure, or death. Local failure-free survival (LFFS) was defined as the freedom from LF and death from any cause. The Kaplan-Meier method was used to estimate the time to events and analyzed via log-rank. Statistical comparisons to test differences between groups were made with the Wilcoxon Signed Rank test. The pre-determined threshold for statistical significance was $p < 0.05$.

### Reporting summary
Further information on research design is available in the Nature Portfolio Reporting Summary linked to this article.

## Results

### Patient and tumor characteristics
Baseline patient, tumor, and treatment characteristics are summarized in Table 1. Patients had a median age of 65.5 years, were mostly female (64.3%), and had an Eastern Cooperative Oncology Group (ECOG) baseline performance status of 1 (57.1%). Nine patients (64.3%) had non-metastatic primary lung cancers that mainly consisted of either adenocarcinoma (28.6%) or squamous cell carcinoma (28.6%) histologies, and five patients (35.7%) had metastatic lesions. All lesions met either the HILUS[7] A (85.7%) or B (14.3%) grouping definitions for ultracentral tumors. No tumors demonstrated any evidence of bronchial invasion but four (28.6%) were directly abutting. The two group B tumors (14.3%) were directly abutting intermediate bronchi and were 1.1 cm and 1.2 cm from the main bronchus. Two patients (14.3%) had tumors that were local recurrences from prior radiation therapy, and two others (14.3%) had thoracic radiation therapy for other unrelated malignancies. Only one patient (7.1%) had any concurrent systemic therapy. The median daily adaptive workflow time was 20.4 min (13.9–45.5 min) per fraction. The median treatment delivery time was 15.5 min (12.0–25.25 min) per fraction.

### Clinical outcomes and toxicities
The median follow-up was 17.15 months (0.5–40.31 months). Two patients were lost to follow-up, but all other patients were either still being actively followed or died at the time of analysis. The 2-year LC, LFFS, OS, and PFS rates were 92.9%, 85.7%, 92.9%, and 64.3%, respectively (Figs. 2 and 3). The median was not reached for any clinical outcome. The single patient who experienced an LF had stage IVB neuroendocrine carcinoma and a pre-treatment maximum diameter of 4.8 cm.

The toxicities associated with SMART are summarized in Table 2. Toxicities were mild, with only three (21.4%) patients experiencing acute grade 2 toxicity and one (7.1%) experiencing late grade 2 toxicity. There were no grade 3, 4, or 5 toxicities associated with SMART observed at any time point.

### Per fraction dosimetry analyses
A total of 112 fractions were delivered, and the adaptive workflow was triggered and performed 90% ($n = 101$) of the time. Weighted optimizations and full readaptations accounted for 30% ($n = 30/101$) and 70% ($n = 71/101$) of the total plan adaptations, respectively. The adaptation workflow was triggered for dosimetric PBT, esophagus, and target coverage violations 26 (23%), 13 (12%), and 39 (35%) times, respectively. The remainder of the adapted fractions were for vessel or spinal cord violations. Three predicted plans were not able to be generated from the ViewRay TPS. The per fraction medians and IQR were generally similar across the base, delivered, and predicted per fraction plans (Supplementary Table 2).

The dosimetric analysis of delivered fractions versus predicted fractions is summarized in Table 3. When PBT violation triggered plan adaptation, SMART demonstrated improvements over the predicted non-adaptive plans for PBT $D_{Max}$ (62.1 Gy vs. 64.9 Gy; $p < 0.001$), $D_{0.03cc}$ (61.0 Gy vs. 63.4 Gy; $p < 0.001$), and $D_{0.1cc}$ (58.2 Gy vs. 61.7 Gy; $p < 0.001$). Additionally, when target under coverage triggered plan adaptation, SMART demonstrated improvements in target coverage for GTV $D_{Min}$ (53.1 Gy vs. 50.9 Gy; $p < 0.001$), GTV $D_{Max}$ (72.6 Gy vs. 74.7 Gy; $p = 0.009$),

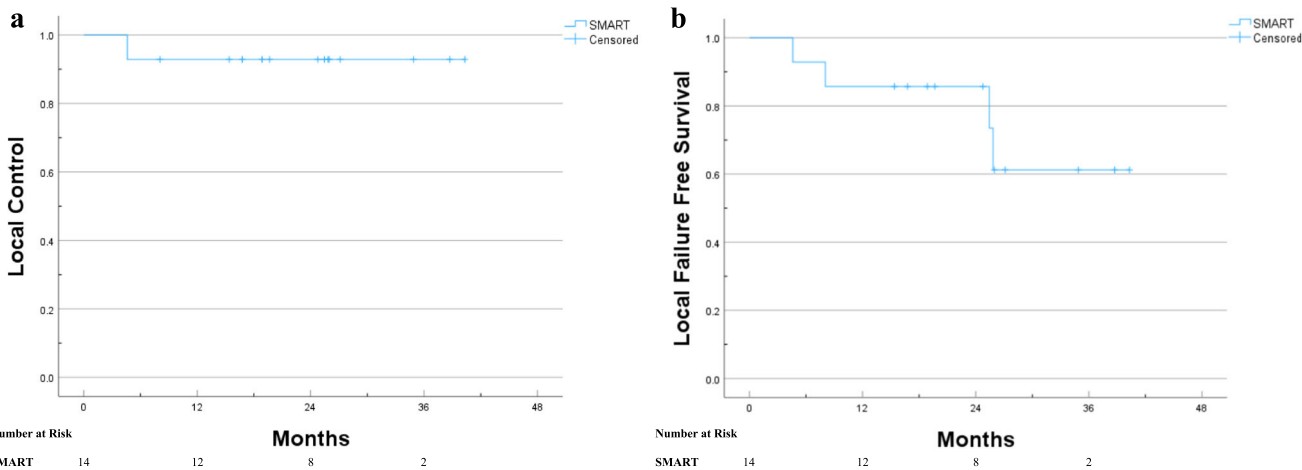

**Fig. 2 | Kaplan–Meier curves for LC of the ultracentral lung lesions and LFFS of the ultracentral lung lesion patients treated with SMART. a** Kaplan–Meier curve of the LC of the ultracentral lung lesions treated with SMART. **b** Kaplan Meier curve of the LFFS for ultracentral lung patients treated with SMART.

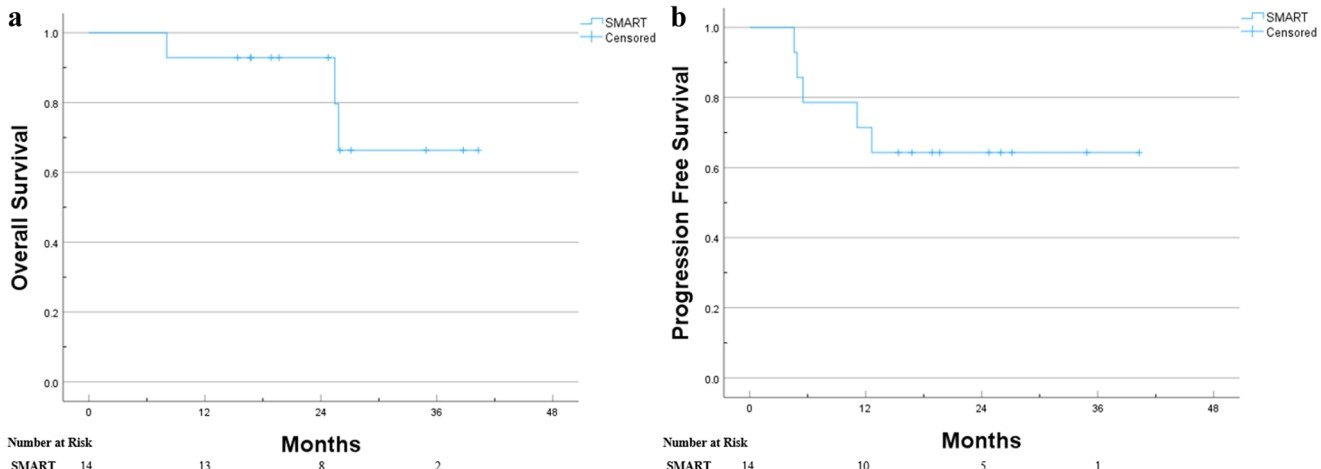

**Fig. 3 | Kaplan–Meier curves for OS and PFS of the ultracentral lung lesion patients treated with SMART. a** Kaplan–Meier curve of the OS for ultracentral lung patients treated with SMART. **b** Kaplan–Meier curve of the PFS for ultracentral lung patients treated with SMART.

GTV $D_{100\%}$ (95.7% vs. 93.1%; $p < 0.001$), GTV $D_{95\%}$ (99.6% vs. 97.4%; $p < 0.001$), GTV $D_{0.03cc}$ (72.1 Gy vs. 73.9 Gy; $p = 0.022$), GTV $D_{0.1cc}$ (71.4 Gy vs. 72.8 Gy; $p = 0.046$), PTV $D_{Min}$ (36.2 Gy vs. 36.0 Gy; $p < 0.001$), and PTV $D_{95\%}$ (95.5% vs. 91.4%; $p < 0.001$).

Structure-specific RIR-based dosimetric analysis of SMART cumulative delivered doses versus cumulative predicted doses are summarized in Table 4. SMART demonstrated improvements over the cumulative predicted doses for both sparing of the PBT and target coverage metrics. SMART had improvements in the PBT dosimetric parameters of $D_{Max}$ (59.4 Gy vs. 65.1 Gy; $p = 0.002$), $D_{0.03cc}$ (56.7 Gy vs. 63.1 Gy; $p = 0.002$), and $D_{0.1cc}$ (55.9 Gy vs. 60.6 Gy; $p = 0.008$). SMART also had better target coverage parameters for GTV $D_{100\%}$ (99.1% vs. 91.8%; $p = 0.002$), GTV $D_{95\%}$ (99.9% vs. 97.2%; $p = 0.003$), PTV $D_{95\%}$ (99.3% vs. 89.3%; $p < 0.001$), and PTV $D_{Min}$ (52.1 Gy vs. 36.2 Gy; $p = 0.008$).

## Discussion

MR-guided daily plan adaptation feasibility has been shown for central lung lesions[11,27,34]. However, in this study, we aimed to investigate the dosimetric impact of daily plan adaptation of SMART for ultracentral lung lesions that are at the highest risk for treatment-related death: HILUS group A and B lesions[7]. Our results demonstrated that daily plan adaptation in SMART reliably improved target coverage and reduced doses to the proximal airways. This means that despite the specific reasons for daily plan adaptation

trigger (e.g., OAR dose violation), the cumulative dose delivered over the course of therapy resulted in improvements for both target coverage and OAR dose violation when compared to predicted cumulative doses for non-adapted therapy. The delivered treatment resulted in a low rate of acute and late toxicities, with only three (21.4%) patients experiencing an acute grade 2 toxicity and one (7.1%) experiencing a late grade 2 toxicity; no grade 3 or higher toxicities were observed at any time point. SMART demonstrated excellent 2-year LC and LFFS rates of 92.9% and 85.7%, respectively. Cumulative dose analysis with structure-specific RIR demonstrated that SMART was associated with median improvements in target coverage and the clinically critical dosimetric parameters of PBT $D_{Max}$, $D_{0.03cc}$, and $D_{0.1cc}$ by 5.7 Gy, 3.6 Gy, and 4.7 Gy, respectively. The cumulative delivered doses align with the excellent toxicity profile and disease control associated with SMART within the cohort. Additionally, the RIR cumulative dose analysis revealed improvements in all dosimetric parameters for the delivered plans as compared to per fraction, suggesting that RIR analysis for daily adaptive therapy may better represent the true dosimetric advantages of SMART than per fractional analysis alone.

Ablative and near-ablative doses delivered to the proximal airway can result in life-threatening complications, such as bronchopulmonary hemorrhage[5,7,9,41–43]. Long-term outcomes of SABR for central and ultracentral lung estimate that the actuarial 5-year rate of severe treatment-related toxicity (i.e., grade 3 or greater) approaches 35%, and 89% of these

## Table 2 | Acute and late toxicities associated with SMART

| Toxicity | Acute | | Late | |
|---|---|---|---|---|
| | Grade 2 n (%) | Grades 3–5 n (%) | Grade 2 n (%) | Grades 3–5 n (%) |
| Thoracic pain | 0 | 0 | 0 | 0 |
| Cough, NOS | 0 | 0 | 0 | 0 |
| Dyspnea, NOS | 1 (7.1%) | 0 | 1 (7.1%) | 0 |
| Odynophagia | 1 (7.1%) | 0 | 0 | 0 |
| Dysphagia | 0 | 0 | 0 | 0 |
| COPD exacerbation | 1 (7.1%) | 0 | 0 | 0 |
| Pneumonitis | 0 | 0 | 0 | 0 |
| Pneumothorax | 0 | 0 | 0 | 0 |
| Fistula | 0 | 0 | 0 | 0 |
| Pulmonary fibrosis | 0 | 0 | 0 | 0 |
| Bronchopulmonary hemorrhage | 0 | 0 | 0 | 0 |

*COPD* chronic obstructive pulmonary disease, *NOS* not otherwise specified, *ROI* region of interest, *SMART* stereotactic magnetic resonance image-guided adaptive radiation therapy.

## Table 3 | Per fraction delivered plans versus predicted plans

| ROI dosimetric parameters | Median (IQR) | | Δp-Value[a] |
|---|---|---|---|
| | Delivered | Predicted | |
| **All fractions (n = 109)** | | | |
| *PBT* | | | |
| $D_{Max}$ (Gy) | 61.8 (59.7–62.9) | 61.0 (57.3–63.1) | 0.204 |
| $D_{0.03cc}$ (Gy) | 60.7 (58.5–61.7) | 59.8 (55.7–62.2) | 0.218 |
| $D_{0.1cc}$ (Gy) | 58.0 (53.9–60.4) | 58.0 (51.5–60.5) | 0.089 |
| *Esophagus* | | | |
| $D_{Max}$ (Gy) | 27.69 (16.4–37.9) | 25.2 (16.1–36.9) | **0.005** |
| $D_{0.03cc}$ (Gy) | 27.0 (15.9–36.6) | 24.5 (15.8–35.5) | **<0.001** |
| $D_{0.1cc}$ (Gy) | 26.3 (15.1–34.9) | 23.8 (15–33.3) | **<0.001** |
| *GTV* | | | |
| $D_{Min}$ (%) | 54.3 (44.0–58.8) | 53.2 (39.26–58.54) | **<0.001** |
| $D_{Max}$ (%) | 72.7 (70.8–78.3) | 77.8 (73.37–80.94) | **<0.001** |
| $D_{100\%}$ (Gy) | 96.4% (92.0–99.6%) | 96.4% (91.1–99.2%) | **0.004** |
| $D_{95\%}$ (Gy) | 99.8% (97.4–100%) | 99.5% (96.1–100%) | **<0.001** |
| $D_{0.03cc}$ (Gy) | 72.1 (70.4–77.8) | 77.2 (72.3–80.1) | **<0.001** |
| $D_{0.1cc}$ (Gy) | 71.6 (69.7–77.3) | 76.0 (71.5–78.9) | **<0.001** |
| *PTV* | | | |
| $D_{Min}$ | 42.9 (30.6–54.4) | 41.3 (27.0–53.4) | **<0.001** |
| $D_{95\%}$ (Gy) | 97.5% (90.5–99.4%) | 95.1% (88.9–98.5%) | **<0.001** |
| **PBT violation (n = 26)** | | | |
| *PBT* | | | |
| $D_{Max}$ (Gy) | 62.1 (60.6–63.0) | 64.9 (62.9–67.1) | **<0.001** |
| $D_{0.03cc}$ (Gy) | 61.0 (59.3–61.7) | 63.4 (61.5–66.0) | **<0.001** |
| $D_{0.1cc}$ (Gy) | 58.2 (57.3–60.4) | 61.7 (59.4–63.3) | **<0.001** |
| **Esophageal violation (n = 12)** | | | |
| *Esophagus* | | | |
| $D_{Max}$ (Gy) | 40.2 (39.1–41.1) | 41.3 (40.7–46.1) | 0.099 |
| $D_{0.03cc}$ (Gy) | 38.7 (38.1–39.7) | 40.1 (39.7–43.7) | 0.108 |
| $D_{0.1cc}$ (Gy) | 36.8 (35.4–38.0) | 38.8 (37.7–39.6) | 0.108 |
| **Target coverage violation (n = 38)** | | | |
| *GTV* | | | |
| $D_{Min}$ (Gy) | 53.1 (47.1–57.1) | 50.9 (44.7–54.7) | **<0.001** |
| $D_{Max}$ (Gy) | 72.6 (70.4–77.8) | 74.7 (72.5–80.9) | **0.009** |
| $D_{100\%}$ (%) | 95.7% (90.2–98.1%) | 93.1% (86.6–97.0%) | **<0.001** |
| $D_{95\%}$ (%) | 99.6% (97.7–100%) | 97.4% (95.4–99.8%) | **<0.001** |
| $D_{0.03cc}$ (Gy) | 72.1 (70.4–77.5) | 73.9 (72.3–80.0) | **0.022** |
| $D_{0.1cc}$ (Gy) | 71.4 (69.8–77.1) | 72.8 (71.8–79.2) | **0.046** |
| *PTV* | | | |
| $D_{Min}$ (Gy) | 36.2 (31.4–48.5) | 36.0 (26.7–46.5) | **<0.001** |
| $D_{95\%}$ (%) | 95.5% (89.1–98.8%) | 91.4% (86.0–96.2%) | **<0.001** |

*GTV* gross tumor volume, *IQR* interquartile range, *PBT* proximal bronchial tree, *PTV* planning target volume, *ROI* region of interest.
[a]Statistically significant *p*-values (i.e., <0.05) are bolded.

bronchial toxicities were comprised of patients with ultracentral lesions[44]. Twelve percent of patients experienced treatment-related death, with most of these deaths (58%) being related to late bronchial hemorrhage[44]. The HILUS trial, a prospective Nordic multicenter phase II study, reported a 15.2% rate of SBRT-related death for the tumors that were the closest to the PBT[7]. These deaths were most often the result of late bronchopulmonary hemorrhage, and 87.5% of those patients had an EQD2 > 100 Gy to their main bronchus. For comparison, the $D_{Max}$ to the PBT within our study demonstrated an EQD2 mean of only 79.2 Gy with no patients' PBT receiving >100 Gy cumulatively. On an expanded analysis, treatment related death was also associated with an elevated dose to the intermediate bronchus; thus, the intermediate bronchus should be considered as part of the mainstem bronchi[45]. Importantly, both direct tumor abutment of the PBT and PBT max dose (including the intermediate bronchus) were found to be associated with treatment-related death on multivariable analysis[45].

MRLs allow for real-time visualization of the tumor and surrounding structures, thus improving target delineation and avoidance of local OARs[17,29,46]. SMART's ability to improve target coverage and reduce doses to the proximal airway effectively widens the therapeutic window for high-risk thoracic tumors[11,30,31]. Perhaps the greatest benefit of SMART for these high-risk tumors is the ability to eliminate the need for an IGTV while maintaining the ability to reliably deliver the prescribed dose to the target via real-time direct tumor localization via cine MRI with gating[29–31]. A recent study of MRgART for central thoracic oligometastatic disease reported a 1-year local control rate of 94% with no grade three or higher acute or late toxicities observed[5]. Those results are consistent with our study's findings. However, what this study sought to evaluate in greater depth is the dosimetric impact of daily plan adaptation. To further investigate the dosimetric benefits of daily plan adaptation effect on SMART's dosimetry, we employed a structure-specific RIR method, which has not been previously reported to the best of our knowledge, to determine a more accurate cumulative dose delivered estimate over the course of SMART for ultracentral lesions.

Determining the true cumulative dose delivered is difficult[35]. A common technique is the use of deformable image registration (DIR). DIR presents several challenges that can affect the accuracy of dose calculations, but no cumulative dose approach is without limitations[38]. Firstly, DIR is highly dependent on the quality of input images, which can be affected by factors such as noise, artifacts, and low resolution. Poor image quality often leads to poor image registration between different image sets, subsequently resulting in inaccurate dose accumulation[36,37,47]. Additionally, the choice of registration algorithm can impact the accuracy of DIR, as various algorithms have different assumptions, sensitivities, and robustness that can contribute

to uncertainties in the registration process[37,47]. Secondly, the complexity of the deformation field in DIR can lead to errors, particularly in ROI with low image contrast or poorly defined anatomical structures. These errors related to the mapping of movement and the deformation of such structures throughout the course of treatment lead to inaccuracies within the cumulative dose[36,37,47]. Consequently, this has a direct impact on the evaluation of the delivered dose and, ultimately, its association with clinical outcomes.

These limitations of DIR make alternative methods like RIR necessary. Structure-specific RIR can overcome many of the limitations of DIR by separately analyzing ROI without relying on deformation fields. This technique reduces uncertainties in dose accumulation by avoiding dose warping and offers a more accurate representation of the cumulative dose delivered to ROI. However, structure-specific RIR also has some limitations. RIR is ultimately dependent on the user's judgment as to when ROIs are properly aligned, only one ROI can be properly analyzed at a time to generate the most accurate cumulative dose. The value of this approach may also diminish based on structure rigidity (e.g., this approach is particularly suitable for the PBT but may have decreased accuracy for a deformable structure such as the esophagus) and geometric complexity (due to the increased difficulty of performing an accurate image registration). Finally, structure-specific RIR analysis is time-consuming and is not currently feasible to perform within the adaptive workflow, limiting its use to retrospective analyses.

The application of RIR to estimate dose accumulation may help to explain the increased therapeutic ratio associated with MRgART[30,31,46,48]. The cumulative treatment courses demonstrated a considerable dosimetric advantage in favor of SMART for ultracentral lung lesions. Delivered plans enhanced target coverage and protected the PBTs in comparison to the predicted treatment course. Figure 4 demonstrates how daily plan adaptation carves out high isodose lines from two intermediate bronchi that sandwich a group B lesion. The degree of this difference seen on RIR cumulative dose analysis (Table 4) was notably greater than per fraction analysis (Table 3). There are likely two factors that contribute to this improvement: (1) feathering of high and low dose spots with each adaptive delivered plan and (2) the treating physician's knowledge of prior delivered plans influencing each successive adaptive plan's coverage decisions. The geometric difference within each adapted plan creates varied regions of high and low doses as compared to a static plan delivered for all fractions, which serves to restrict cumulative hotspots but also increases dose minimums. This effect can be seen in the decrease of the cumulative delivered plan PBT hotspots and the increase of the dose minimums to the PTV (increased PTV $D_{Min}$ of 15.9 Gy). This effect also likely contributes to the improved target coverage over the course of treatment. The other important contributing factor that is elucidated with cumulative plan analysis and not as easily discernable on per fraction analysis are the treatment decisions for individual ROI dose violations. What cannot be captured with dosimetric parameter data is the actual geometric location of hotspots within each ROI. However, the treating physician knows the geometric locations of both high and low dose spots when visually evaluating the dose distribution each day, which therefore allows them to make decisions to accept either lower target coverage or higher PBT doses with the understanding of how each successive fraction is being delivered. This could possibly explain why there did not appear to be any differences between the delivered and predicted fractions (Table 3) for PBT hotspots, but substantial differences between the delivered and predicted fractions were illustrated by the cumulative plans.

Additional limitations of this study include inherent biases due to its retrospective nature, such as selection bias and underreporting of toxicities. The study cohort is relatively small, and a sample size analysis was not performed prior to data collection. Important clinical outcomes of the cohort did not reach statistical significance due to the low number of events within our follow-up time. Toxicities observed within the cohort had both a low event rate and grade, and thus, it was not possible to evaluate for clinically meaningful correlations between SMART dosimetry and specific toxicities. As with all daily online adaptive MRgRT, there are inherent dose

**Table 4 | Analysis of RIR cumulative delivered plans versus cumulative predicted plans**

| ROI dosimetric parameters | Median (IQR) | | Δp-Value[a] |
|---|---|---|---|
| | **Delivered** | **Predicted** | |
| *Proximal bronchial tree (n = 14)* | | | |
| $D_{Max}$ (Gy) | 59.4 (57.0–60.2) | 65.1 (62.3–66.2) | **0.002** |
| $D_{0.03cc}$ (Gy) | 56.7 (54.8–58.6) | 63.1 (60.3–64.6) | **0.002** |
| $D_{0.1cc}$ (Gy) | 55.9 (48.3–58.6) | 60.6 (58.0–63.4) | **0.008** |
| Target (n = 14) | | | |
| *GTV* | | | |
| $D_{Min}$ (Gy) | 55.9 (48.0–59.8) | 54.9 (48.2–58.1) | 0.084 |
| $D_{Max}$ (Gy) | 70.0 (68.9–75.8) | 71.3 (70.0–77.1) | 0.345 |
| $D_{100\%}$ (%) | 99.1% (94.5–100%) | 91.8% (86.4–96.6%) | **0.002** |
| $D_{95\%}$ (%) | 99.9% (97.6–100%) | 97.2% (92.4–99.1%) | **0.003** |
| $D_{0.03cc}$ (Gy) | 69.9 (68.1–75.5) | 73.5 (70.2–76.4) | 0.117 |
| $D_{0.1cc}$ (Gy) | 69.7 (67.6–75.1) | 72.8 (69.6–75.9) | 0.084 |
| *PTV* | | | |
| $D_{Min}$ (Gy) | 52.1 (34.2–57.2) | 36.2 (29.3–51.2) | **0.008** |
| $D_{95\%}$ (%) | 99.3% (86.4–99.9%) | 89.3% (80.6–95.8%) | **<0.001** |

*GTV* gross tumor volume, *IQR* interquartile range, *PBT* proximal bronchial tree, *PTV* planning target volume, *ROI* region of interest.
[a]Statistically significant *p*-values (i.e., <0.05) are bolded.

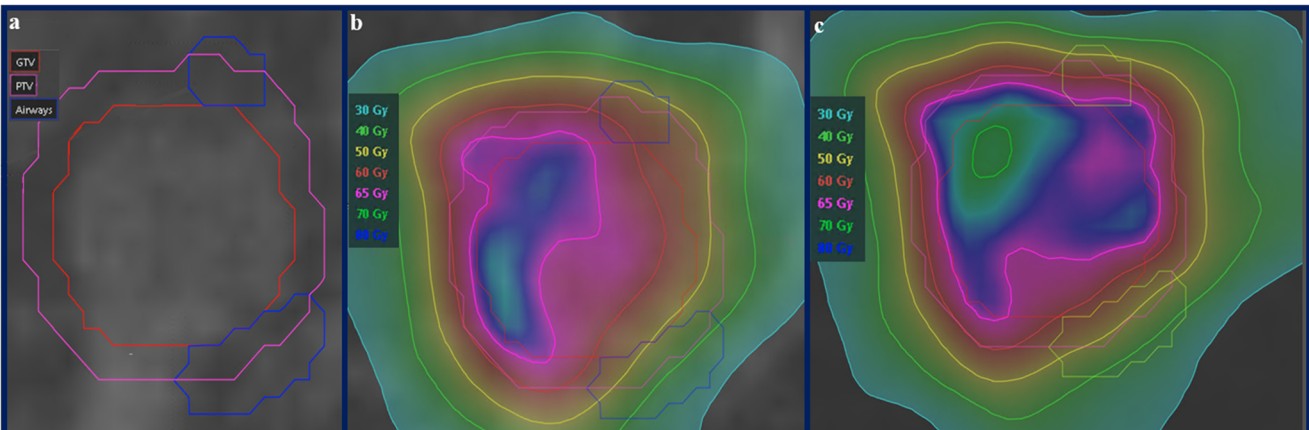

**Fig. 4 | Example of the contours, cumulative delivered dose, and cumulative predicted dose for the highest risk axial slice where the PBT directly abutted and sandwiched a HILUS group B lesion that underwent online daily adaptation for seven out of eight total fractions. a** GTV (red), PTV (pink), and PBT (blue) contours. **b** The cumulative delivered dose plan demonstrates how daily plan adaptation pushes the higher isodose lines outside of the proximal airway while maintaining excellent target coverage. **c** The cumulative predicted dose plan demonstrates doses nearing 80 Gy within the thin-walled PBT.

uncertainties due to the use of a deformed planning CT. Accounting for the high degree of air–tissue interface within the lung adds to the difficulty of accurately predicting dose distributions using this method. In addition, every patient was treated within our department, which was an early adopter of SMART for ultracentral lung lesions. Thus, these safety data may not translate to centers with less experience in SMART for ultracentral lung lesions. Nevertheless, our cohort represents one of the largest reports on the use of SMART for ultracentral lesions and the first to perform structure-specific RIR cumulative dose analysis to evaluate the true dosimetric impact of daily adaptation over the course of therapy.

## Conclusion

SMART demonstrated dosimetric advantages and excellent clinical outcomes for HILUS ultracentral lung tumors. Cumulative dose analysis using a structure-specific RIR approach demonstrated that daily plan adaptation with SMART reliably improved target coverage and reduced doses to the proximal airways simultaneously and helped further characterize the improved therapeutic window seen with SMART. This structure specific-specific RIR approach provided insights that elucidate the dosimetric advantages of SMART more so than per fractional analysis alone. Prospective studies are needed to validate the clinical outcomes of this study.

## Data availability

Source data for the figures are available as Supplementary Data 1. Individual-level medical imaging data cannot be provided to protect patient privacy.

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

## Author contributions

J.M.B. contributed to project conception, data collection, data analysis, primary paper writing, and paper editing; R.C.C. contributed to data collection and paper editing; A.G. contributed to data collection and paper editing; C.L. contributed to data analysis and manuscript editing; J.W. contributed to data analysis and paper editing; A.N. contributed to data collection and paper editing; E.K. contributed to paper editing; M.L.S. contributed to data collection and paper editing; A.S. contributed to data collection and paper editing; B.P. contributed to paper editing; T.J.D. contributed to paper editing; G.R. contributed to paper editing; J.A. contributed to paper editing; L.N. contributed to paper editing; A.O.G. contributed to paper editing; V.F. contributed to paper editing; K.L. contributed to data collection and paper editing; S.A.R. contributed to project conception, data analysis, and paper editing.

## Competing interests

K.L. and V.F. declare the following competing interests: both received consulting fees from ViewRay, Inc. S.A.R. declares the following competing interests: has been the recipient of research grants, has received an honorarium, and has served on the Lung Research Consortium Advisory Board for ViewRay, Inc. T.D. declares the following competing interests: holds equity shares in Moderna and has received consulting fees from AstraZeneca. Furthermore, he has received both consulting and sponsored travel fees from the NCCN. The other authors declare no competing interests.

## Additional information

**Article**

**J.M. Bryant** [1] ✉**, Ruben J. Cruz-Chamorro**[1]**, Alberic Gan**[2]**, Casey Liveringhouse**[1]**, Joseph Weygand**[1]**, Ann Nguyen** [2]**, Emily Keit**[1]**, Maria L. Sandoval**[1]**, Austin J. Sim** [3]**, Bradford A. Perez**[1]**, Thomas J. Dilling** [1]**, Gage Redler**[1]**, Jacqueline Andreozzi**[1]**, Louis Nardella**[1]**, Arash O. Naghavi**[1]**, Vladimir Feygelman**[1]**, Kujtim Latifi**[1] **& Stephen A. Rosenberg**[1] ✉

[1]Department of Radiation Oncology; H. Lee Moffitt Cancer Center & Research Institute, Tampa, FL, USA. [2]University of South Florida Health Morsani College of Medicine, Tampa, FL, USA. [3]Department of Radiation Oncology; James Cancer Hospital, Ohio State University Comprehensive Cancer Center, Columbus, OH, USA. ✉e-mail: john.bryant@moffitt.org; Stephen.Rosenberg@moffitt.org

