## [Peer review File · Communications Medicine]

Structure-Specific Rigid Dose Accumulation Dosimetric Analysis of Ablative Stereotactic MRI-Guided Adaptive Radiation Therapy in Ultracentral Lung LesionsReviewers' comments:

Reviewer #1 (Remarks to the Author):

this is small retrospective series (14 patients over 2 years) of a heterogeneous population of patients (lung primary and metastases) who were treated with stereotactic body radiotherapy technique for ultra central disease using a novel device that allows real time volumetric imagin during treatment and adaptive planning

Radiation using SBRT in the lung for central and ultra central tumors have been known to have an increase in morbidity and mortality which has previously been shown to be related to dose and dose per fraction, tumor size, heterogeneity of dose and proximity of the tumor to the central airways

this study had a clinical component - which focused on safety although it is difficult to learn a lot from this experience given its size - smaller than a phase I trial for radiation. Results seem great but almost unbelievable especially when you show relative small changes in the replanning potion of the study

the second part of the study was a dosimetric analysis using a novel approach to replanning and comparison

these would my suggestions to improve the manuscript

1. how were patients selected o be treated with the MRI linac during this time span for an ultracental tumor or were ALL patients treated this way. only 30% of lung cancer pts could undergo DIBH

2. were these pts smokers or non-smokers? given the use of DIBH

3. visualization of mass on CT vs MR - was this always concordant - what would you do if it was not?

4. how many breath holds were required to complete treatment ? Imaging was used to verify 3 mm was adequate for PTV

5. on day of treatment - how long did it take to assess plan, modify plan, did you ever repeat imaging after the modification and find a need to adapt again ? or this was not done. how many breath holds and how long was treatment?

6. Cumulative incidence of local control with death as competing risk should be calculated given many had metastatic disease. Toxicity data is dependent on follow up time and time alive. How reliably were follow ups collected and was anyone lost to follow up? that data should be available. People could die at OSH from hemorrhage and not be documented in the EMR. also there is discrepancy in the follow up - mean follow up of 37 months but figures very few pts at 3 yrs - only 2 of 14. you should use median follow up rather mean (should not be potential follow up either). based on the follow up seen in your curves re: numbers at risk you should report 18 to 24 month outcomes

7. table 1 - suggest tumor size not GTV size, suggest smoking hx

8 differences in dose seem to small - do this could explain differences in complications or is it complete related to smaller volumes

the planning technique used for central lesions - do you use a homogeneous approach or do you try to use hotspots in the GTV?

a figure showing the tumor on the MRI could be helpful if zoomed out to show comparison to OAR adjacent to tumor

Reviewer #2 (Remarks to the Author):

The authors present an overall very well-written and thorough manuscript discussing an important topic due to oftentimes lethal toxicity: SBRT treatment of ultra-central lung tumors. The authors should be proud of their work, as the paper was well-constructed, sound, and easy to understand.

The manuscript reports dosimetric and toxicity outcomes for a cohort of 14 consecutive patients receiving SMART (stereotactic MRI guided adaptive radiotherapy) for ultra-central lung tumors, utilizing a 60 Gy in 8 fractions schema. Delivered plans, either the base plan recalculated on daily anatomy or plans adapted to daily anatomy, were compared to predicted plans, or base plans recalculated on daily anatomy. Small but often statistically significant differences were observed between individual delivered

and predicted fractions. However, large differences in cumulative dose were observed between predicted and delivered plans. Importantly, the authors cleverly avoided the major pitfalls of both global deformable and rigid reconstructions by performing two local rigid registrations: one for the GTV and one for the closest part of the proximal bronchial tree (PBT). This enabled a more precise estimation of cumulative target and OAR dose since the GTV registration was only used for target evaluation and the PBT registration was only used for OAR evaluation. Lastly, good local control, overall survival, and progression free survival was observed, as evidenced by Kaplan Meir analysis.

The methods, results, and discussion are generally quite sound; my minor comments are listed below.

1. Very minor point, line 55 should be: "...analysis to a single structure at a time."
2. Lines 78-81 require more detail for somebody who doesn't specialize in SMART to follow, in my opinion. How was the structure used for gating contoured and determined? There could be many candidates inside the PTV, correct? Does percentage excursion threshold <5% mean that the beam was turned off once less than 95% of the gating structure was within its 3mm margin? I would suggest adding detail that clarifies these sentences to a non-expert.
3. Lines 90-92: GTVs were rigidly registered and edited if needed, but this was not common. Was this because there was not sufficient time for the tumor to respond to RT with so few fractions? Was tumor growth observed? I would add to this sentence why it was not common.
4. Line 92 says that the prior plan was either the base plan or the previous adaptive plan recalculated on daily anatomy. This is only mentioned once to my memory, but its implications should be clarified. Are the base plan and prior adaptive plan both visible, and selectable, in the SMART workflow even if daily adaption is not an option because all constraints were met? If so, aren't some delivered plans adapted to the previous day's anatomy, making them neither truly adaptive nor base plans re-calculated on that day's anatomy? The presence of treating with a prior adaptive plan should be further clarified.
5. Line 94: Plans were only re-optimized if violations occurred. How long did full treatments take to deliver, and approximately how long did optimization take? I understand that this is a vendor specified workflow, but I would add discussion concerning if potential adaptive benefit was lost when fractions were treated with a recalculated plan, rather than re-optimized? How does the cost of extra patient time spent on the couch compare to the potential benefit of always optimizing to daily anatomy. In Ethos, for instance, the predicted plan and optimized plan are performed for every fraction, and the superior plan is chosen. In essence, what is the cost of not performing both by default?
6. Results section, subsection Clinical outcomes and Toxicities: you do not cite Figure 2 anywhere in the text; this should be corrected.
7. The only issue I found with data presentation is that mean values are presented instead of medians. Were data tested for normality, for example with the Shapiro Wilk's test? The Wilcoxon signed rank is for nonparametric data in which skew is assumed. Means are heavily effected by skew, and medians should be therefore be used with asymmetric data. The authors should justify their use of the Wilcoxon test, either by saying they tested for normality or assume the data to be non-parametric. Medians should then be used in tables instead of means, as means should be reserved for normal data.
8. Line 282 should read: "Perhaps the greatest benefit of SMART..."

9. More detail should be given in the methods about how registrations were performed? Were they performed by a single physician or physicist? Was scripting was involved? Were the GTVs aligned to the tumor centroid or to the surface closest to the PBT? Even if it was a manual and subjective process, it should be described to the best of the author's ability, as this is a key component of the manuscript.

10. An inherent added uncertainty in all online adaptive platforms is reliability of synthetic CT (or deformed planning CT due to daily anatomy) dose calculation. This implications of this added uncertainty should be discussed for the MRIdian adaptive platform and this treatment site.

Dear Reviewers,

Thank you for your additional suggestions for our manuscript and we are deeply appreciative of your time. We have sought to address your comments thoroughly. In doing so, we believe that we have improved the clarity of our statements, included greater detail of our processes, and enhanced the readability of the manuscript overall. We appreciate your ongoing consideration. Please find direct replies to your comments below.

Reviewers' comments:

Reviewer #1 (Remarks to the Author):

This is small retrospective series (14 patients over 2 years) of a heterogeneous population of patients (lung primary and metastases) who were treated with stereotactic body radiotherapy technique for ultra central disease using a novel device that allows real time volumetric imaging during treatment and adaptive planning.

Radiation using SBRT in the lung for central and ultra central tumors have been known to have an increase in morbidity and mortality which has previously been shown to be related to dose and dose per fraction, tumor size, heterogeneity of dose and proximity of the tumor to the central airways. This study had a clinical component - which focused on safety although it is difficult to learn a lot from this experience given its size - smaller than a phase I trial for radiation. Results seem great but almost unbelievable especially when you show relatively small changes in the replanning portion of the study. The second part of the study was a dosimetric analysis using a novel approach to replanning and comparison.

These would my suggestions to improve the manuscript:

1. how were patients selected to be treated with the MRI linac during this time span for an ultracentral tumor or were ALL patients treated this way. Only 30% of lung cancer pts usually undergo DIBH.

All patients on the MR-Linac are treated this way. Due to the gating capabilities of the MRIdian system, even patients who had higher difficulty with breath holds could successfully complete the treatment although it would often take longer to deliver. We thank you for highlighting the importance of patient selection criteria in our study.

2. were these pts smokers or non-smokers? given the use of DIBH

I have added these details within table 1.

3. visualization of mass on CT vs MR - was this always concordant - what would you do if it was not?

Thank you for this important question. We always found the CT and MR concordant. The tumor and normal anatomy as seen on the MRIdian TRUFI sequence is what were used to define the geometries of all planning structures. The lower field strength (0.35 T) of the MRIdian suffers from less geometric distortion as compared to conventional field strength (1.5-3 T) of most diagnostic MRIs; therefore, significant differences between CT and the TRUFI sequence would not be expected. We appreciate your attention to the concordance between CT and MR imaging

4. how many breath holds were required to complete treatment ? Imaging was used to verify 3 mm was adequate for PTV.

We do not have data regarding the total amount of breath holds; however, we have added treatment time data to give additional insight. Although patients were encouraged to perform DIBH during therapy, the unit would

automatically beam-off when the target moved outside of the pre-determined gating structure. This enabled safe treatment delivery even with patients who had difficulty with breath hold. The tracking structure was kept inside of the 3 mm envelope, so the PTV expansion of 3 mm is reasonable. The sum of the square of the errors of uncertainty in geometric distortion, MR to linac isocenter differential and other variables in set up is only 1-2 mm (PMID 30706022). In that context, 3 mm margin accounts for all these errors. The length of time that a patient could hold their breath did not affect the 3 mm boundary that was set for all patients.

5. On day of treatment - how long did it take to assess plan, modify plan, did you ever repeat imaging after the modification and find a need to adapt again ? or this was not done. how many breath holds and how long was treatment?

Thank you for your thoughtful inquiry about our daily workflow. We have added the daily workflow times. Repeat imaging with a TRUFI sequence was not performed after adaptive planning; however, the target and the surrounding OARs are visualized in during treatment with the real-time tracking and gating capabilities of the MRIdian unit. Please see our response to question 4 regarding DIBHs.

6. Cumulative incidence of local control with death as competing risk should be calculated given many had metastatic disease. Toxicity data is dependent on follow up time and time alive. How reliably were follow ups collected and was anyone lost to follow up? that data should be available. People could die at OSH from hemorrhage and not be documented in the EMR. Also, there is discrepancy in the follow up - mean follow up of 37 months but figures very few pts at 3 yrs - only 2 of 14. you should use median follow up rather mean (should not be potential follow up either). based on the follow up seen in your curves re: numbers at risk you should report 18-to-24-month outcomes.

You bring up important points and we have made all these edits. Thank you.

7. For Table 1, I suggest using tumor size rather than GTV size and adding smoking history details.

We agree and have added smoking history details and changed GTV to tumor size. Your suggestion to include smoking history details was indeed valuable.

8. Differences in dose seem to small - could this explain the differences in complications or is it related to smaller volumes?

The differences in dose are related to the subtle daily anatomic differences seen at time of setup. The treatment volumes are roughly the same because the base plans were made with the intention of treating with real-time tumor tracking and beam gating. We only have the data within this cohort for patients treated with daily on-line adaptive therapy, so we are unable to make an inference regarding the toxicity profile differences if the predicted doses had been delivered. However, the greatest risk factor for serious toxicity is related to the max dose at the PBT and esophagus, and we believe that the dosimetric improvements are clinically meaningful.

9. The planning technique used for central lesions - do you use a homogeneous approach or do you try to use hotspots in the GTV?

We allow up to a 120% hotspot within the GTV. We have added this language within the text. Additional planning details can be found within the supplementary table 1. We are thankful for your suggestion to help us better explain the planning technique.

10. A figure showing the tumor on the MRI could be helpful if zoomed out to show comparison to OAR adjacent to tumor.

We agree that a figure that gives a greater view of these UC lesions in relation to their OARs would be beneficial. We have added this figure to the manuscript. Thank you for this helpful suggestion!

Reviewer #2 (Remarks to the Author):

The authors present an overall very well-written and thorough manuscript discussing an important topic due to oftentimes lethal toxicity: SBRT treatment of ultra-central lung tumors. The authors should be proud of their work, as the paper was well-constructed, sound, and easy to understand.

The manuscript reports dosimetric and toxicity outcomes for a cohort of 14 consecutive patients receiving SMART (stereotactic MRI guided adaptive radiotherapy) for ultra-central lung tumors, utilizing a 60 Gy in 8 fractions schema. Delivered plans, either the base plan recalculated on daily anatomy or plans adapted to daily anatomy, were compared to predicted plans, or base plans recalculated on daily anatomy. Small but often statistically significant differences were observed between individual delivered and predicted fractions. However, large differences in cumulative dose were observed between predicted and delivered plans. Importantly, the authors cleverly avoided the major pitfalls of both global deformable and rigid reconstructions by performing two local rigid registrations: one for the GTV and one for the closest part of the proximal bronchial tree (PBT). This enabled a more precise estimation of cumulative target and OAR dose since the GTV registration was only used for target evaluation and the PBT registration was only used for OAR evaluation. Lastly, good local control, overall survival, and progression free survival was observed, as evidenced by Kaplan Meir analysis.

The methods, results, and discussion are generally quite sound; my minor comments are listed below.

1. Very minor point, line 55 should be: "...analysis to a single structure at a time."

Thank you for finding this error! We have fixed it.

2. Lines 78-81 require more detail for somebody who doesn't specialize in SMART to follow, in my opinion. How was the structure used for gating contoured and determined? There could be many candidates inside the PTV, correct? Does percentage excursion threshold <5% mean that the beam was turned off once less than 95% of the gating structure was within its 3mm margin? I would suggest adding detail that clarifies these sentences to a non-expert.

The tumor itself was utilized as the primary structure for gating. This critical component of our procedure was initially contoured by experienced therapists, and subsequently verified for accuracy by a physician. The tumor itself was most often used as the ROI for the gating structures. Furthermore, a threshold of less than 5% means that the beam was turned off whenever less than 95% of the tracking structure pixels of the gating structure remained within its designated 3 mm margin. Thank you for your insightful query regarding these important specifics of our gating procedures and we appreciate your suggestion to elaborate on these details. We have added a more comprehensive explanation to the revised manuscript.

3. Lines 90-92: GTVs were rigidly registered and edited if needed, but this was not common. Was this because there was not sufficient time for the tumor to respond to RT with so few fractions? Was tumor growth observed? I would add to this sentence why it was not common.

This was due to the small degree of geometric changes seen throughout treatment over such a short course of therapy. We have added language to clarify this point. Thank you!

4. Line 92 says that the prior plan was either the base plan or the previous adaptive plan recalculated on daily anatomy. This is only mentioned once to my memory, but its implications should be clarified. Are the base plan and prior adaptive plan both visible, and selectable, in the SMART workflow even if daily adaption is not an option because all constraints were met? If so, aren't some delivered plans adapted to the previous day's anatomy, making them neither truly adaptive nor base plans re-calculated on that day's anatomy? The presence of treating with a prior adaptive plan should be further clarified.

We value your point on the implications of using the prior adaptive plan. The prior adaptive plan is as valid a starting point as a base plan. Either one is recalculated on the today's anatomy to see if the dose distribution corresponding to the fluence produced by that plan is satisfactory or requires adaptation (re-optimization). We have added language to the manuscript to better explain the adaptive workflow.

5. Line 94: Plans were only re-optimized if violations occurred. How long did full treatments take to deliver, and approximately how long did optimization take? I understand that this is a vendor specified workflow, but I would add discussion concerning if potential adaptive benefit was lost when fractions were treated with a recalculated plan, rather than re-optimized? How does the cost of extra patient time spent on the couch compare to the potential benefit of always optimizing to daily anatomy. In Ethos, for instance, the predicted plan and optimized plan are performed for every fraction, and the superior plan is chosen. In essence, what is the cost of not performing both by default?

I have added data to the results section that covers treatment and workflow times. There are no significant differences in time regarding a re-optimization versus full adaptation.

Our institutional protocol was to only adapt when triggers for adaptation are met. If predicted plan satisfied the clinical goals then we have no reason to adapt. There is an argument to be made that this saves some time, and you won't need to re-optimize and QA the predicted plan.

6. Results section, subsection Clinical outcomes and Toxicities: you do not cite Figure 2 anywhere in the text; this should be corrected.

Thank you for pointing out this omission! We have rectified this issue within the revised manuscript.

7. The only issue I found with data presentation is that mean values are presented instead of medians. Were data tested for normality, for example with the Shapiro Wilk's test? The Wilcoxon signed rank is for nonparametric data in which skew is assumed. Means are heavily effected by skew, and medians should be therefore be used with asymmetric data. The authors should justify their use of the Wilcoxon test, either by saying they tested for normality or assume the data to be non-parametric. Medians should then be used in tables instead of means, as means should be reserved for normal data.

After reviewing, we agree with your recommendation. We have updated the tables and text to medians with IQRs instead of means.

8. Line 282 should read: "Perhaps the greatest benefit of SMART...".

Thank you for catching this! We have corrected this within our revised manuscript.

9. More detail should be given in the methods about how registrations were performed? Were they performed by a single physician or physicist? Was scripting was involved? Were the GTVs aligned to the tumor centroid or to the surface closest to the PBT? Even if it was a manual and subjective process, it should be described to the best of the author's ability, as this is a key component of the manuscript.

Registrations were conducted through a carefully estimated and manually iterative process and performed by a single physician. The alignment of the Gross Tumor Volumes (GTVs) was straightforward, with minimal changes in their geometry observed throughout the treatment. For the Proximal Bronchial Trees (PBTs), the alignment focused on the smallest segments in closest proximity to the tumor, particularly those at the highest risk. The process involved aligning the edge nearest to the tumor while ensuring the overall geometry remained consistent with the contiguous structure of the remaining PBT. We have incorporated these critical details into the methods section of our manuscript. We are grateful for your suggestion to elaborate on the registration process, as it adds clarity to our methods.

10. An inherent added uncertainty in all online adaptive platforms is reliability of synthetic CT (or deformed planning CT due to daily anatomy) dose calculation. This implications of this added uncertainty should be discussed for the MRIdian adaptive platform and this treatment site.

We are grateful for your observation regarding the deformed planning CT dose calculation. Thank you for bringing this omission to our attention. We have added language within our limitations paragraph to address this.

REVIEWERS' COMMENTS:

Reviewer #2 (Remarks to the Author):

The authors addressed all of my comments in a satisfactory manner.

Reviewer #3 (Remarks to the Author):

Many thanks for allowing me the opportunity to review this excellent manuscript. I am entering the review process at the second stage and note this. I have also noted the original reviewers comments and the rebuttal both of which will have enhanced the original manuscript for the reader.

The arena of central and ultra central SABR is interesting and of great importance to patients with concerns regarding toxicity. The MRL has a unique role to play and the data here shows its potential utility. We look forward to further data in this arena.

At this stage I am happy with the manuscript as it is and do not require any further amendments.